# The Future Climate under Different CO₂ Emission Scenarios Significantly Influences the Potential Distribution of *Achnatherum inebrians* in China

**Jia-Min Jiang [1], Lei Jin [1], Lei Huang [2] and Wen-Ting Wang [1],***

[1] School of Mathematics and Computer Science, Northwest Minzu University, Lanzhou 730030, China; y201730458@stu.xbmu.edu.cn (J.-M.J.); jinlei9719@163.com (L.J.)
[2] Northwest Institute of Eco-Environment and Resources, Chinese Academy of Sciences, Lanzhou 730000, China; huanglei@lzb.ac.cn
* Correspondence: iamwwt1983@163.com

**Abstract:** The threat posed by poisonous weeds to grassland ecosystems may be exacerbated by climate change mainly driven by carbon dioxide (CO₂) emissions. *Achnatherum inebrians* is a common and poisonous grassland weed that is seriously endangering the sustainable development of prairie animal husbandry in Western China. Understanding the influence of future climate change under different CO₂ emission scenarios on the potential distributions of *A. inebrians* is critical for planning agricultural strategies to manage the continued invasion. An ecological niche model (ENM) was developed using Maxent to predict the potential distribution of *A. inebrians* under three different CO₂ emission scenarios. Occurrence records of *A. inebrians* were selected utilizing the nearest neighbor method. Six environmental variables, which were identified through principal component analysis, correlation analysis and their contribution rates, were used to perform the ENM. At the same time, considering the uncertainties of predicting future climates, four global circulation models were used for the Maxent projections with average results calculated. Our results demonstrate differential influences of various CO₂ emission scenarios on the potential distributions of *A. inebrians*. Before 2050, high CO₂ emission scenarios resulted in a wider potential distribution of *A. inebrians*, when compared to low CO₂ emission scenarios. However, after 2050, the low CO₂ emission scenarios were more conducive to an expanded potential distribution. In addition, after 2050, high CO₂ emission scenarios maintain the geographical distribution centroids of *A. inebrians* in lower latitudes, while low CO₂ emission scenarios result in distribution centroids rising to higher latitudes. Further, low CO₂ emission scenarios resulted in the average potential distribution elevation dropping lower than in high CO₂ emission scenarios.

**Keywords:** ecological niche model; potential distributions; invasive weeds; geographical distribution centroids



## 1. Introduction

Increasing greenhouse gas concentrations are linked to rising global mean sea surface temperatures, alongside climate changes in precipitation patterns, storm severity, and sea level [1–3]. The majority of anthropogenic greenhouse gas (GHG) emissions are carbon dioxide (CO₂) released from burning fossil fuels, resulting in the steady increase in atmospheric concentrations of CO₂ since the onset of the industrial revolution [4]. However, the concentration of CO₂ in the atmosphere is regulated by many natural processes [5], and therefore the prediction of future climates is challenging. To address this uncertainty, the Intergovernmental Panel on Climate Change (IPCC) Fifth Assessment Report (AR5) introduced representative concentration pathways (RCPs), including RCP 2.6, RCP 4.5 and RCP 8.5 that depict climate scenarios in different greenhouse gas emissions [6]. RCP 2.6 represents a future climate with low CO₂ emissions, whereby global annual GHG emissions peak

between 2010 and 2020, after which emissions fall significantly resulting in a 450 ppm $CO_2$ concentration in 2100, and global average temperatures have increased by 0.2–1.8 °C. RCP 4.5 is a medium $CO_2$ emission scenario, with a peak of global annual GHG emissions around 2040, followed by a gradual decline. In RCP 4.5, $CO_2$ concentrations are projected to reach 650 ppm and global average temperatures will increase 1.0–2.6 °C by 2100. RCP 8.5 represents high $CO_2$ emissions, with $CO_2$ emissions continuing to rise throughout the 21st century. $CO_2$ concentrations will increase to 1350 ppm and global average temperatures will increase 2.6–4.8 °C by 2100 [7–9].

Climate change induced by $CO_2$ emissions significantly influences the geographical distributions of plant species worldwide [10]. Changing climates can result in habitat expansion, contraction, and even shifts in plant communities [11–15]. Plant responses to these changing atmospheric conditions are species specific. When 12 European forest tree species were modelled under the future climate (RCP 2.6, 4.6, and 8.5), they were divided into three groups: winners, losers, and alien species. Assuming limited migration, most of these species would face significant reductions in suitable habitat areas as the $CO_2$ emission scenario intensifies [16]. Wróblewska & Mirski (2018) also identified that the geographic range of circumboreal plants will likely decrease in the future, with the extent of the loss directly correlated to $CO_2$ emission scenarios severity [17]. Given the low phenotypic plasticity of weeds, their abundances are also projected to decline concurrent with increasing $CO_2$ concentrations [18]. However, Patterson (1995) found that higher $CO_2$ concentrations can promote photosynthesis and growth in C3 weeds, and improve the water use efficiency in both C3 and C4 weeds [19]. Increasing $CO_2$ emissions can positively influence the distribution and demographics of weeds, and even increase their resistance to herbicides [20–22]. Furthermore, higher levels of atmospheric $CO_2$ could stimulate the growth of some weed species, inducing the production of more tubers and rhizomes in perennial weeds [23–25].

*Achnatherum inebrians* (drunken horse grass), is a perennial herb and a typical grassland poisonous weed. After feeding on it, livestock will experience intoxication such as increased heart rate and staggering gait, and even death [26]. As a result of its increased resistance to environmental extremes, it is widely dispersed and highly adaptable, especially in degraded grasslands [27,28]. Currently, *A. inebrians* is distributed throughout the arid, semiarid, alpine, and subalpine grasslands in Inner Mongolia, Ningxia, Gansu, Xinjiang, Qinghai, and Sichuan of China [29]. Recently the distribution and abundance of *A. inebrians* have been continually increasing, seriously jeopardizing the sustainable development of prairie animal husbandry in Western China [30,31]. Therefore, it is vital for risk estimation and the development of long-term strategies to investigate the potential distribution of *A. inebrians* under future climate change through different $CO_2$ emission scenarios.

Ecological niche models (ENMs) have been frequently used to identify the potential distribution of species following climate change [32–37]. Based on the environmental variables associated with species' occurrence records, ENMs seek to characterize the suitable species-specific environmental conditions, and then identify where they are spatially distributed [38,39]. One of the most popular ENM techniques, the maximum entropy approach (Maxent), estimates species distribution by identifying the probability distribution based on the maximum entropy principle [40,41]. Maxent requires only present records of the species and even functions with small sample sizes by using samples of the background environment [42–44]. However, occurrence data for most species have traditionally been recorded without sufficient supporting documentary information, and can even include errors and bias in geography, resulting in spatial autocorrelation and environmental bias of model simulation [45]. In addition, given the uncertainties of future climatic conditions, it is still challenging to predict the potential distribution of species [46,47]. Future climate conditions are projected from global climate models (GCMs) for different representative concentration pathways (RCPs). Previous studies have combined the parameters of multiple GCMs into ensembles of the GCM projections, in order to reduce the climate uncertainty and produce a more robust and reliable projection [48]. However, this results in a loss of the

spatial patterns produced by each GCMs [10,49]. Differences among various GCMs could be important for understanding and predicting the potential distributions of *A. inebrians*, and thereby developing control strategies.

This study simulated the response of the potential distribution of *A. inebrians* across China to different $CO_2$ emission scenarios, in order to better control its invasion through the following approach: (1) key environmental variables highly correlated with the distribution of *A. inebrians* were identified; (2) a Maxent model was developed for both present and 12 climate change scenarios (4 GCMs×3 RCPs); (3) average results were calculated under three $CO_2$ emission scenarios; (4) analysis of the changes in potential distribution areas of *A. inebrians* after quantification under three $CO_2$ emission scenarios; and (5) the direction of the geographical distribution centroid shifts and average elevation of the potential distribution areas of *A. inebrians* responding to three $CO_2$ emission scenarios were estimated.

## 2. Materials and Methods

### 2.1. Species Occurrence Data

In total, 164 non-overlapping occurrence records of *A. inebrians* in China were collected from the Chinese Virtual Herbarium (http://www.cvh.org.cn/; accessed on 20 January 2019) and Global Biodiversity Information Facility (GBIF Occurrence Download https://doi.org/10.15468/dl.r4t29p; accessed on 20 January 2019). To reduce spatial autocorrelation and avoid over-fitting of our model at intensely sampled locations [50], points that were at 10 km apart from one another and from among the original occurrence data points were chosen, which resulted in 137 occurrences for *A. inebrians*.

### 2.2. Environmental Variables

To construct the ecological niche model (ENM), 19 bioclimatic variables (for the current climate, i.e., the average for the years 1960–1990) of 137 species' occurrence records were first extracted from the corresponding layers using ArcGIS 10. Principal component analysis (PCA) identified important variables where the component matrix was greater than 0.8 in the composition, explaining greater than 80% of the total variability. Finally, bioclimatic variables with weak correlations (r < 0.8) were retained through correlation analysis. The final bioclimatic variables were Bio02, Bio03, Bio06, Bio10, Bio15, Bio16, and Bio19 (Table 1).

**Table 1.** Environmental variables used for ENM to predict the potential future distribution of *A. inebrians*.

| Bioclimatic Variables | Meaning of Variables |
| --- | --- |
| Bio02 | Mean Diurnal Range (Mean of monthly (max temp—min temp)) |
| Bio03 | Isothermality (Mean Diurnal Range/Temperature Annual Range) (×100) |
| Bio06 | Min Temperature of Coldest Month |
| Bio10 | Mean Temperature of Warmest Quarter |
| Bio15 | Precipitation Seasonality (Coefficient of Variation) |
| Bio16 | Precipitation of Wettest Quarter |
| Bio19 | Precipitation of Coldest Quarter |

For the uncertainty of future $CO_2$ emission scenarios, we have adopted three emission scenarios: RCP 2.6, 4.5, and 8.5. For the simulation of future climate under different $CO_2$ emission scenarios, we considered four GCMs: GISS-E2-R (GS), HadGEM2-AO (HD), MIROC5 (MC), and NorESM1-M (NO; detail in Table 2). Based on the dynamic characteristics of the three $CO_2$ emission scenarios, the influences of two future time periods, 2050 (average for 2041–2060) and 2070 (average for 2061–2080), on the potential distributions of *A. inebrians* were analyzed. All environmental data were downloaded from the WorldClim Dataset (http://www.worldclim.com/) with 2.5 arc-min spatial resolution.

**Table 2.** Four GCMs of future climate used to predict the potential future distribution of *A. inebrians*.

| GCM | Code | Institution |
|---|---|---|
| GISS-E2-R | GS | NASA Goddard Institute for Space Studies |
| HadGEM2-AO | HD | National Institute of Meteorological Research/Korea Meteorological Administration |
| MIROC5 | MC | Atmosphere and Ocean Research Institute (The University of Tokyo), National Institute for Environmental Studies, and Japan Agency for Marine-Earth Science and Technology |
| NorESM1-M | NO | Norwegian Climate Centre |

### 2.3. Ecological Niche Model

ENM of *A. inebrians* were generated using Maxent 3.3.3k [40]. Auto features (linear, quadratic, product, and hinge) were set due to our small sample sizes. The regularization parameter was set to 1, and 6000 background points were extracted randomly from the whole territory of China. Model validation was performed using cross-validation procedures with 20 independent replicates. Relative contributions of the environmental variables to the Maxent model were considered in choosing the environmental variables again. After removing the variables with the lowest contributions, the final results were obtained through cross-validation procedures with 20 replicates again. Model performances were evaluated by calculating the area under the curve (AUC) of the receiver operating characteristic plot. AUC values range between 0.5 and 1.0, where a value of 0.5 means model discrimination power is not better than the random and above 0.5 indicates a performance better than the random. The best-performing model for the current scenario was used to project the potential distributions of *A. inebrians* under climate change scenarios. Additionally, the average results are the mean of the potential distributions of *A. inebrians* under 4 GCMs.

The method of the highest sum of sensitivity (true positive rate) and specificity (true negative rate) was used to calculate the threshold (TH) between predicted absenteeism and presence. The potential distributions were manually classified into no adaptive region (<TH), adaptive region (TH-0.7), and high adaptive region (>0.7) by ArcGIS 10. Furthermore, the threshold was used to convert the potential distribution probability into binary, representing the presence and absence of *A. inebrians*. Changes in the distribution areas of 2050 were compared to current distribution, and those of 2070 were compared to 2050, respectively.

### 2.4. Data Analysis

It was assumed that the study area was a homogeneous plane and the point at which the species is distributed on the plane where the moment reaches equilibrium is the geographical distribution centroid of the species. The trajectory of the geographical distribution centroid of a species over a period of time can reflect the general trend of the distribution of the species. The study area was two-dimensionally meshed according to the resolution of $2.5'$, i.e., 5 m × 5 m. Then, the geographical distribution centroid was calculated in accordance with the following formula:

$$N = \frac{\sum\limits_{j=1}^{m} N_i \times P_{i,j}}{\sum\limits_{i=1}^{n} \sum\limits_{j=1}^{m} P_{i,j}} \tag{1}$$

$$E = \frac{\sum\limits_{i=1}^{n} P_{i,j} \times E_j}{\sum\limits_{i=1}^{n} \sum\limits_{j=1}^{m} P_{i,j}} \tag{2}$$

where $P_{i,j}$ is the potential distribution probability of *A. inebrians* in the area $(i, j)$, $N_i$ and $E_j$ are the latitude and longitude of the area $(i, j)$, and $N$ and $E$ are the latitude and longitude of the geographical distribution centroid.

The average elevation of the potential distributions was calculated as follows:

$$E_{avg} = \frac{\sum\limits_{i=1}^{n} E_{i,j} \times P_{i,j}}{\sum\limits_{i=1}^{n} \sum\limits_{j=1}^{m} P_{i,j}} \tag{3}$$

where $E_{i,j}$ is the elevation of the area $(i, j)$, and $E_{avg}$ is the average elevation of the potential distributions.

## 3. Results

### 3.1. Model Performance and Importance of Predictor Variables

The contributions of seven environmental variables: Bio02, Bio03, Bio06, Bio10, Bio15 Bio16, and Bio19 were 1%, 9.5%, 15.1%, 17.4%, 14.5%, 16.9%, and 25.6%, respectively. By removing Bio02, the Maxent model of *A. inebrians* had a higher predictive power, such that the AUC = 0.91 ± 0.05 (mean ± SD) was increased by 0.01. When re-analyzed the contributions of the six environmental variables of Bio03, Bio06, Bio10, Bio15 Bio16, and Bio19 were 5.5%, 15.1%, 16.2%, 13.5%, 16.8%, and 25.9%.

The potential distribution probability of suitable habitats for *A. inebrians* can be maintained at a high level, the range of which varies slightly between 0.53 and 0.59, when the isothermality is between 30 and 45 (Figure 1a). In addition, the potential distribution probability exhibits a hump curve with increased temperature and precipitation (Figure 1b–f). When the minimum temperature of the coldest month equaled 12.05 °C the potential distribution probability reached its peak value (Figure 1b). The response curves also show that the suitable precipitation seasonality range is between 90.9 and 98.1, and that the potential distribution probability of *A. inebrians* exceeds 0.6 (Figure 1d). Similarly, the potential distribution probability rapidly reaches 0.6 when the precipitation of the wettest quarter increases to 218 mm, then rapidly decreases once the precipitation of the wettest quarter exceeds 300 mm (Figure 1e). The potential distribution probability is higher with a lower volume of precipitation during the coldest quarter (Figure 1f).

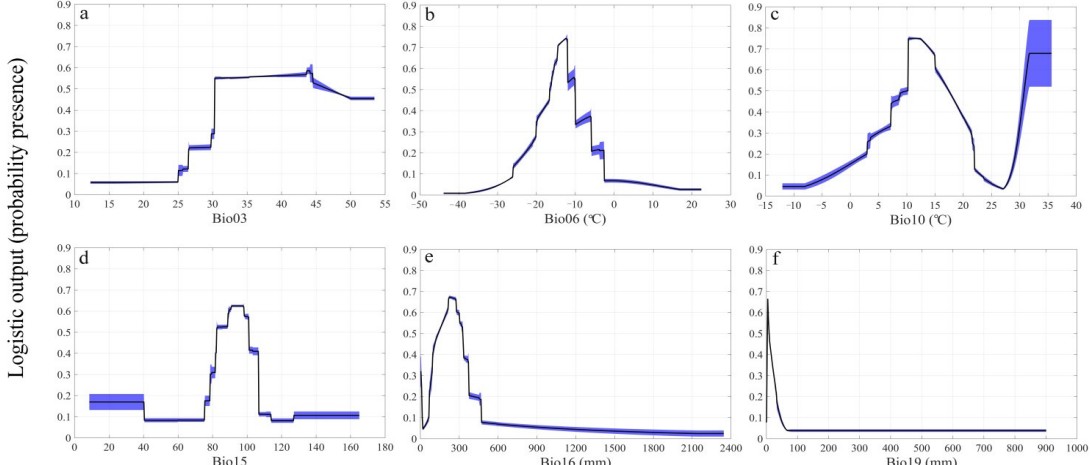

**Figure 1.** Response curves display the relationships between the potential distribution probability of *A. inebrians* and six environmental variables, including (**a**) Isothermality (Bio03), (**b**) Min Temperature of Coldest Month (Bio06), (**c**) Mean Temperature of Warmest Quarter (Bio10), (**d**) Precipitation Seasonality (Bio15), (**e**) Precipitation of Wettest Quarter (Bio16), and (**f**) Precipitation of Coldest Quarter (Bio19). Values shown are the average over 20 replicate runs; blue margins show ±SD calculated over 20 replicates.

### 3.2. The Influence of CO2 Emission Scenarios on the Potential Future Distributions of A. inebrians

The potential distributions of A. inebrians under current climatic conditions are classi­fied according to the threshold value of 0.37 (Figure 2). The highly adaptive regions are mainly concentrated in the southwest of Gansu and east of Qinghai, while the adaptive regions are mainly distributed in the southeast of Gansu, Ningxia and north of Shaanxi. Both regions are considered typical temperate grasslands. In addition, the alpine meadow areas are scattered with a number of adaptive regions, such as Western Sichuan, Eastern Tibet and sporadic adaptation zones in Xinjiang.

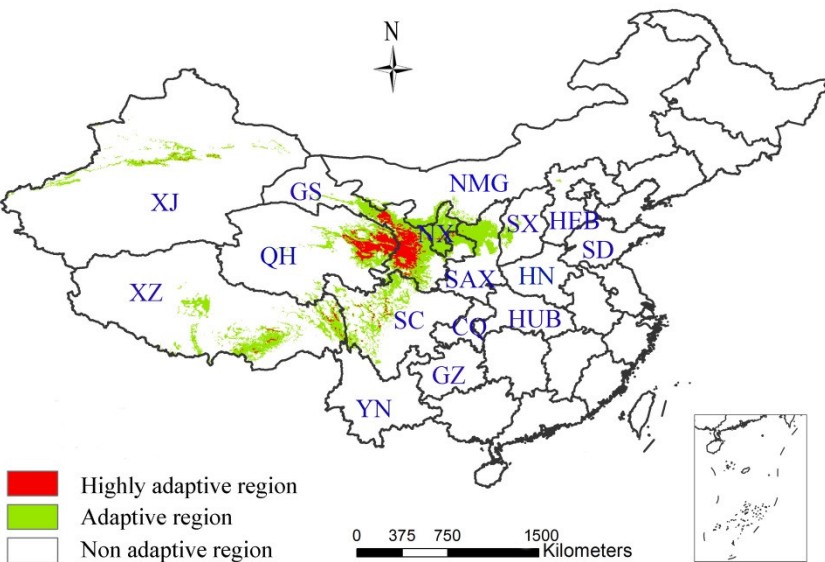

**Figure 2.** The potential distributions of *A. inebrians* under current climatic conditions.

$CO_2$ emission scenarios will continue to promote the gradual expansion of adaptive regions of *A. inebrians* into the future. In 2050, the adaptive region is projected to have expanded southwest (i.e., into the alpine meadow area) and northeast (i.e., into the tem­perate grassland area) with southern Gansu as its center (Figure 3). In the GS and NO models, the expansion characteristics of the adaptive regions are similar, in that as $CO_2$ emission scenarios increase, the area of the adaptive region grows, although the range of the adaptive region is larger in the GS model (Figure 3 and Figure 5a). However, the HD model predicts the exact opposite, indicating that low $CO_2$ emission scenarios are more suitable for the growth of *A. inebrians* (Figure 3 and Figure 5a). The MC model reveals that the adaptive region under high $CO_2$ emission scenarios is larger than with low $CO_2$ emission scenarios, but the adaptive region under medium $CO_2$ emission scenarios is the smallest of all the three (Figure 3 and Figure 5a). In summary, the average results indicate that higher $CO_2$ emission scenarios will cause a wider distribution of the adaptive region of *A. inebrians* by 2050.

After 2050, most of the adaptive regions of *A. inebrians* are stable. With low $CO_2$ emission scenarios, the adaptive regions are expanding, while they retract with high $CO_2$ emission scenarios in all models with the exception of HD. The average results also show that the area of the adaptive regions will have a greater expansion under low $CO_2$ emission scenarios than under high $CO_2$ emission scenarios after 2050 (Figure 4). With the exception of HD, the average data forecast after 2050 shows that the low $CO_2$ emission scenarios are more conducive to the survival of *A. inebrians* (Figure 5b).

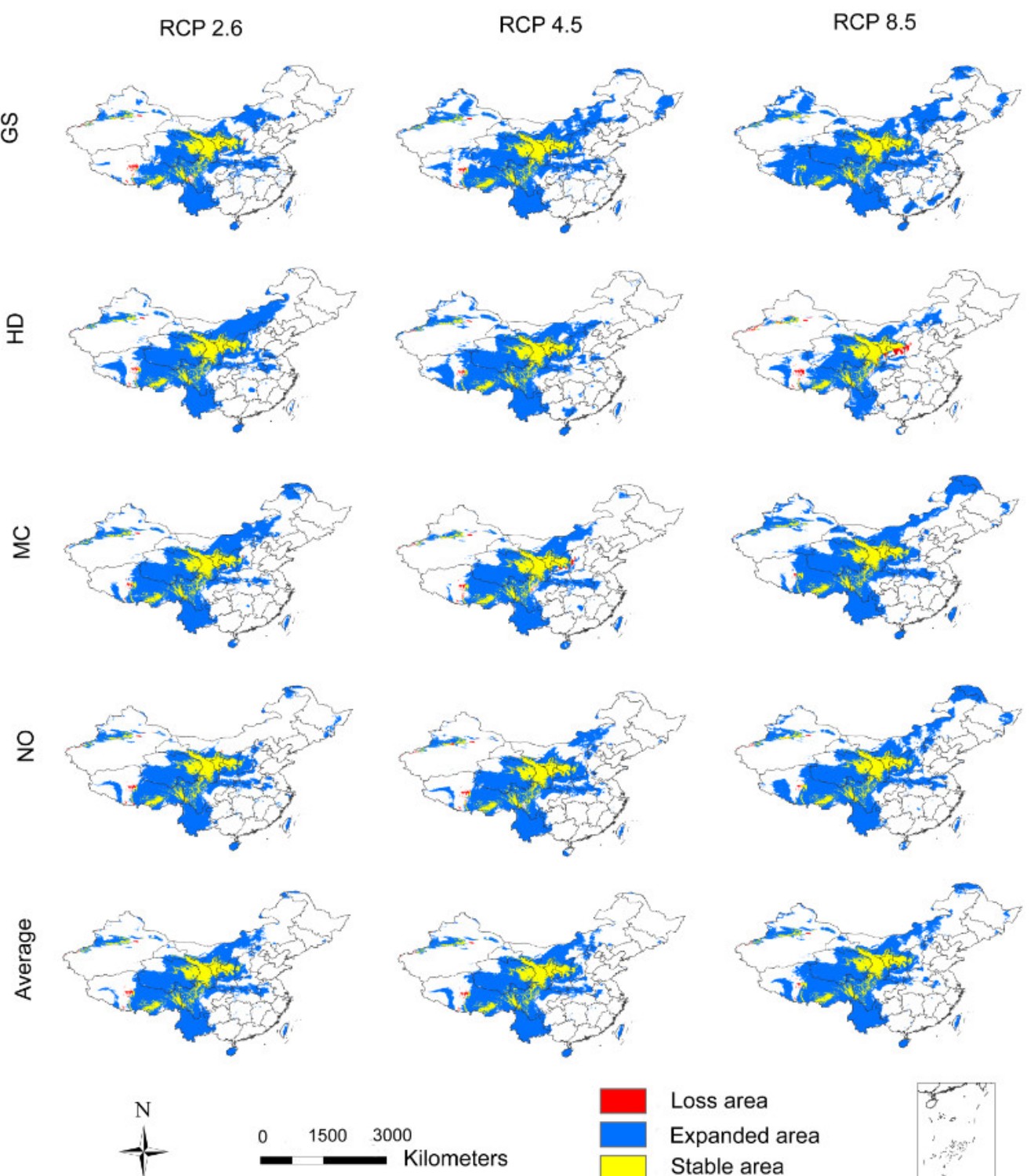

**Figure 3.** The potential distribution changes of *A. inebrians* in 2050 in comparison to current trends. The first four rows are the results of four GCMs, i.e., GS, HD, MC, and NO. Additionally, the last row is the average of the results of four GCMs. The columns show results under three $CO_2$ emission scenarios, i.e., RCP 2.6, 4.5, and 8.5.

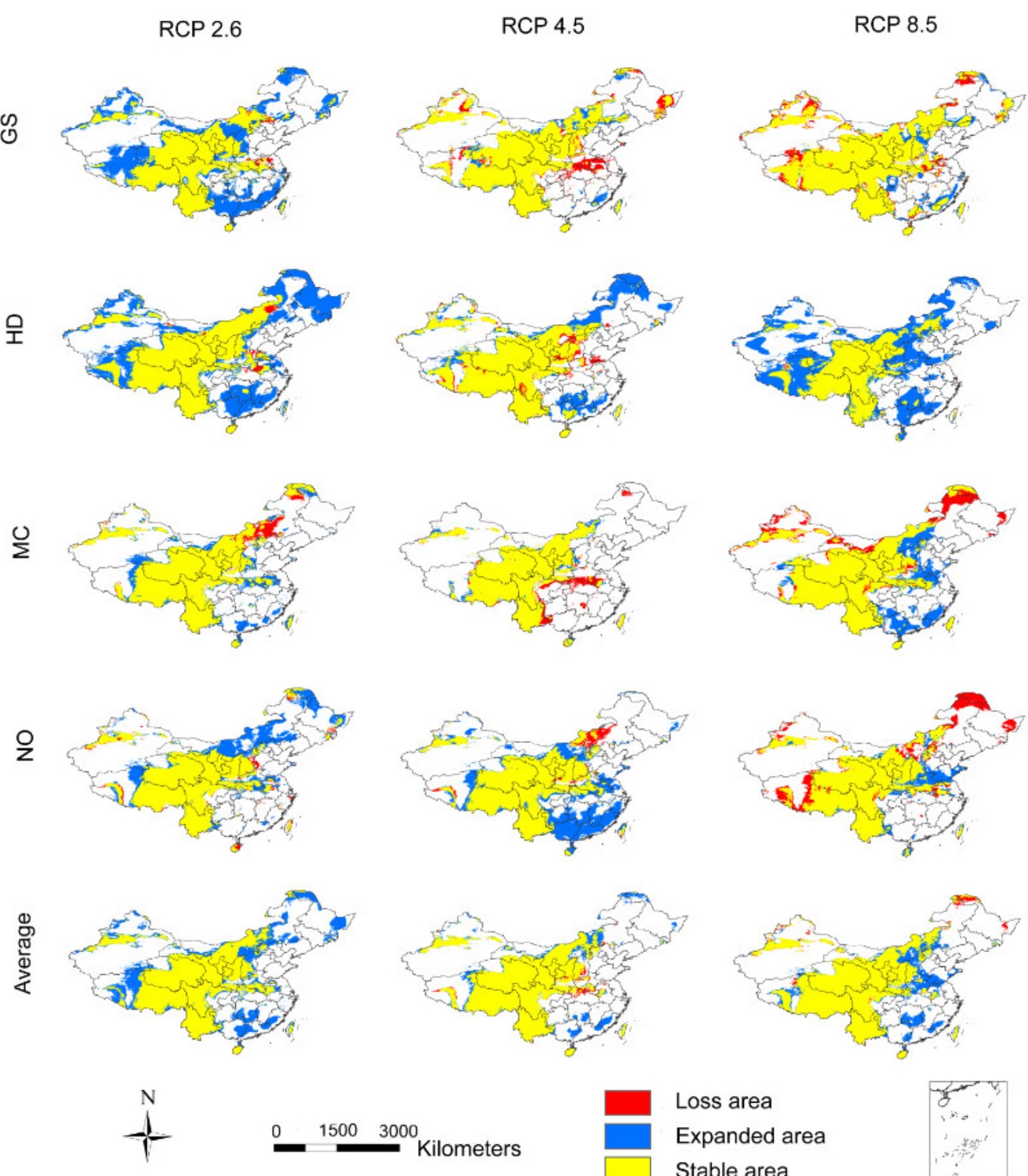

**Figure 4.** The potential distribution changes of *A. inebrians* in 2070 in comparison to 2050. The first four rows are the results of four GCMs, i.e., GS, HD, MC, and NO. Additionally, the last row is the average of the results of four GCMs. The columns show results under three $CO_2$ emission scenarios, i.e., RCP 2.6, 4.5, and 8.5.

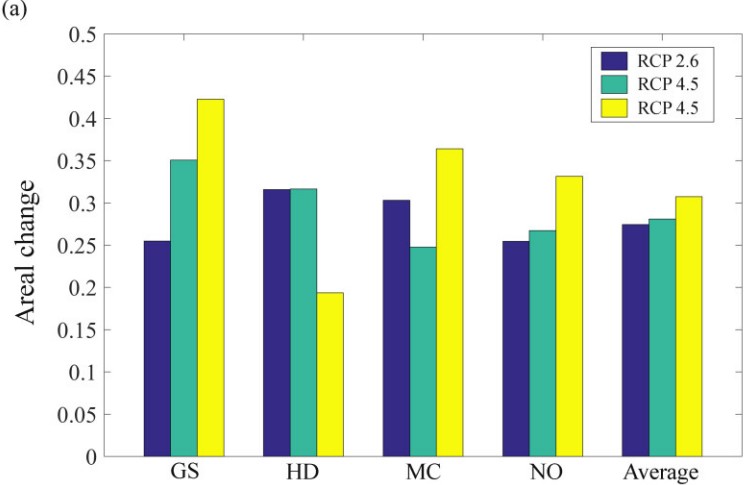

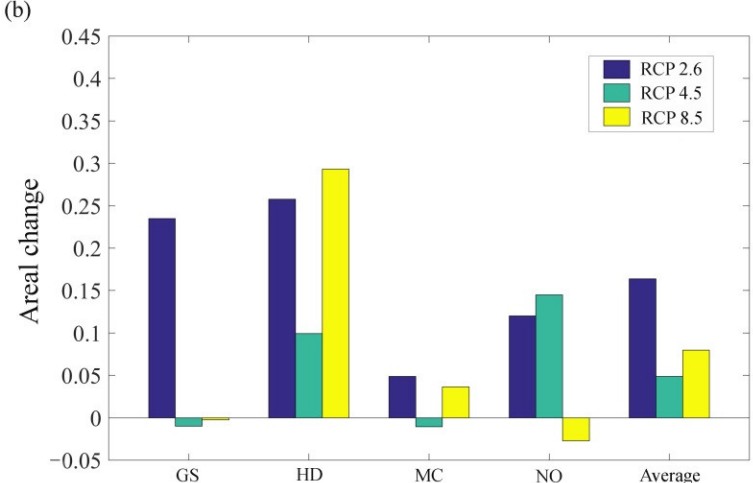

**Figure 5.** The impact of three $CO_2$ emission scenarios (RCP 2.6, 4.5, and 8.5) on the potential distribution changes of *A. inebrians*. (**a**) The changes of the potential distribution in 2050 compared to current trends; (**b**) the changes of the potential distribution in 2070 compared to 2050. GS, HD, MC, and NO are four GCMs, and average represents the average of the results of four GCMs.

### 3.3. The Influence of $CO_2$ Emission Scenarios on the Geographical Distribution Centroid and Average Elevation of the Adaptive Regions of A. inebrians

From current conditions through to 2050, climate changes under the influence of $CO_2$ emission scenarios will likely cause the geographical distribution centroid of *A. inebrians* to move southeast, with a decrease in its latitude (Figure 6a–e). The GS and NO models predict that low $CO_2$ emission scenarios result in a latitudinal decrease in the geographical distribution centroid, whereas HD and MC models predict an increase (Figure 6a–d). The average results show that medium $CO_2$ emission scenarios also result in a latitudinal decrease in the geographical distribution centroid (Figure 6e). However, after 2050, the situation has reversed. With the exception of the MC models, the three others all project that low $CO_2$ emission scenarios can increase the latitude of the geographical distribution centroid, while high $CO_2$ emission scenarios result in a decrease (Figure 6a–e).

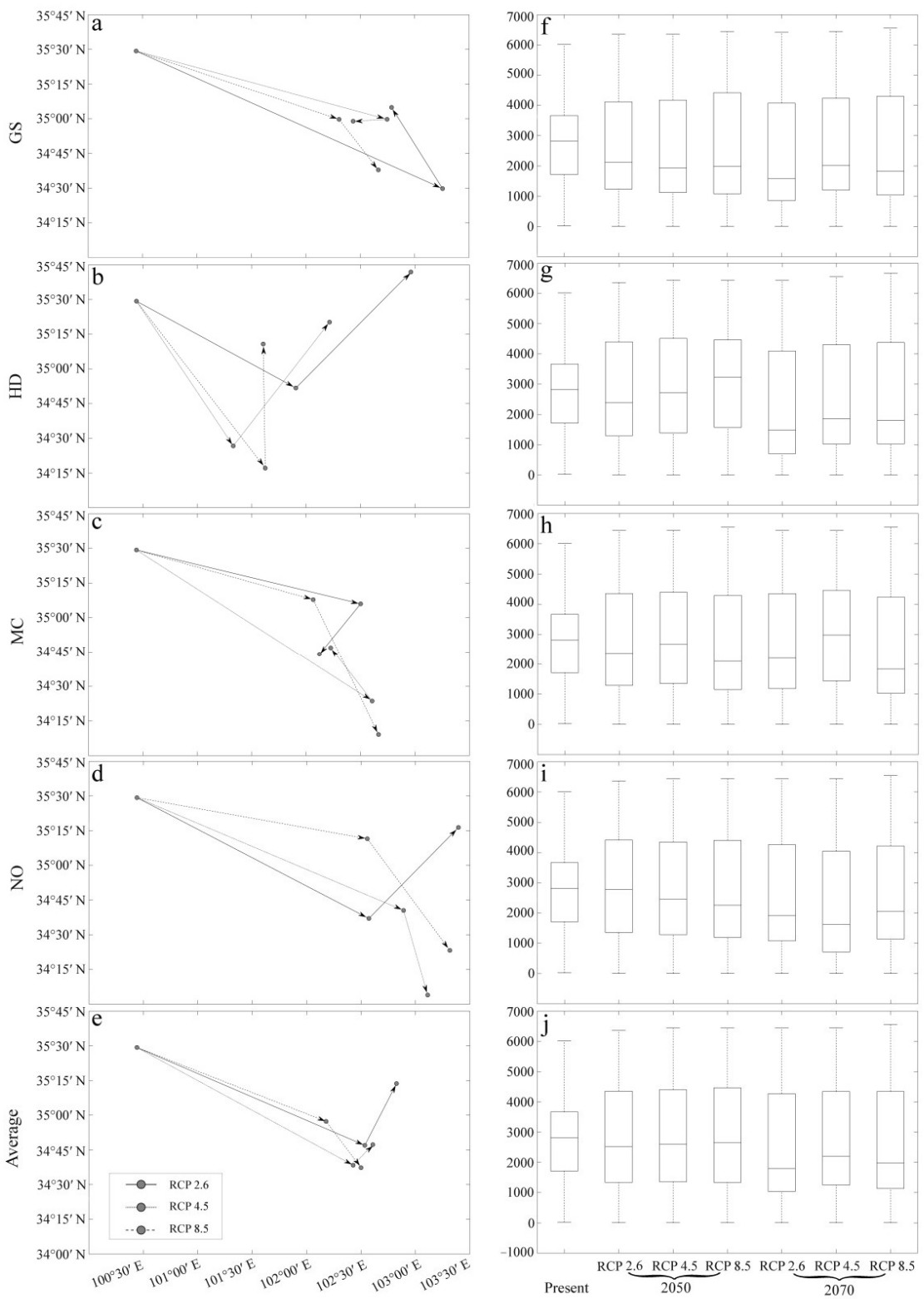

**Figure 6.** The effect of three $CO_2$ emission scenarios (RCP 2.6, 4.5, and 8.5) on the geographical distribution centroid and average elevation of the adaptive regions of A. inebrians. The first four rows are the results of four GCMs, i.e., GS, HD, MC, and NO. Additionally, the last row is the average of the results of four GCMs. (**a–e**) Changes in the geographical distribution centroid of the adaptive regions of *A. inebrians* from present to future (2050 and 2070). The black dot is the geographical distribution centroid, and the arrow represents the direction of time change. (**f–j**) Boxplots of elevation of the adaptive regions of *A. inebrians* under different climate scenarios.

The average elevation of the adaptive regions of *A. inebrians* under the influence of $CO_2$ emission scenarios has a general downward trend. Low $CO_2$ emission scenarios continually decrease the average altitude, but the medium and high $CO_2$ emissions only reveal a trend of lowering the average elevation after 2050 (Figure 6f–i). The average results show that the average elevation of the adaptive regions' decline slows with the increase in $CO_2$ emission scenarios (Figure 6j).

## 4. Discussions

Carbon dioxide ($CO_2$) is the most important greenhouse gas released as a result of anthropogenic activity. This study has modelled the effect of three different $CO_2$ emission scenarios (RCP 2.6, 4.5, and 8.5) on the potential future distributions of *A. inebrians*. The response of the Maxent results to the environmental variables indicates that in the coldest month/quarter, which is also the dormant period of seeds of *A. inebrians*, the potential distribution probability of *A. inebrians* is higher when the minimum temperature and precipitation are lower. This is likely because the seed germination rate of weeds is higher after dormancy in lower temperatures [51–53]. Moreover, light drought stress is more conducive to the embryonic root growth of *A. inebrians* [54]. During the growing season, moderate temperature and rainfall are clearly beneficial to the growth of weeds; hence the ecological niche model also includes two other important factors: the mean temperature of the warmest quarter and the precipitation of the wettest quarter. The increase in $CO_2$ emission concentration has had a significant impact on increasing temperatures in most areas of China, especially in the northwest [55]. In addition, it has influenced the precipitation patterns, with Northwest China becoming even drier and the coastal areas more humid [56]. Furthermore, with increases in $CO_2$ emission concentration, seasonal fluctuations of extreme climates are likely to occur more frequently and with larger amplitudes [15,57].

It is predicted that the suitable regions for *A. inebrians* in 2050 will greatly expand, extending to the Inner Mongolia grassland and the Qinghai–Tibet Plateau, while the expansion range is relatively smaller from 2050 to 2070. Our research supports the conclusions of Saebø and Mortensen (1998) and Singh et al. (2011) that increasing $CO_2$ emission scenarios are beneficial to the growth of perennial herbs [23,24]. However, the reason for the expansion of suitable habitats for *A. inebrians* after 2050 is not clear. It is possible that after 2050, in addition to RCP 8.5, the $CO_2$ emission concentration of other scenarios may be alleviated, especially with the $CO_2$ emission concentration of RCP 2.6 beginning to decline. Additionally, the two time periods we studied were different in length, 50 years and 20 years, respectively. Our research also identified that various intensities of $CO_2$ emissions induce extremely different effects on the expansion of *A. inebrians*. Most GCM (except HD) simulations show that the high $CO_2$ emission scenarios model increase range expansion before 2050; while after 2050, the low $CO_2$ emissions scenarios model results in range expansion. The average results not only draw the same conclusions, but also reveal that the scope of expansion increases with the increase in $CO_2$ emission scenarios before 2050. Our findings are different compared to Dyderski et al. (2018) and Wróblewska & Mirski (2018), as different species have different niches and naturally respond differently to climate change [16,17]. Unlike tree species and circumboreal plants, $CO_2$ may have a positive effect on the growth and reproduction of *A. inebrians*.

Under anthropogenically induced climate change, migration and diffusion have become a significant response mechanism for plants. Many species will disperse to areas with the most suitable climate for their growth to maintain homeostasis. Some studies have found that global warming led to a poleward and upward shift in the range of many plants [13,58,59], but not all plants, as some engaged in southerly migration [60]. The geographical distribution centroids of *A. inebrians* were generally projected to move southeast under different $CO_2$ emission scenarios. However, the direction of the geographical distribution centroids will likely be diversified after 2050, especially under low $CO_2$ emission scenarios with a latitudinal recovery of the geographical distribution centroids. With

the increase in $CO_2$ concentration, there was a predicted decline in the average elevation of the potential distributions. In all GCM models, we identified that the changes in the geographical distribution centroids and average elevation predicted by the HD model were significantly different from the other three. The HD results show that the latitude of the geographical distribution centroids under the low emission scenarios in 2070 was higher than that of the current latitude, and even that under the high $CO_2$ emission scenarios in 2050. This result seems to support the conclusion that plants migrate to higher altitudes and higher latitudes in future climate change scenarios [13,58,59]. Therefore, the impact of $CO_2$ emission scenarios on the potential distribution of *A. inebrians* is strongly influenced by the choice of GCMs.

The uncertainty of future climates is one of the critical issues in accurately predicting the effects of climate change. It is therefore one of the core issues that needs to be addressed for conservation planning of livestock management [10,61]. In this study, four GCMs were used to explore the effect of climate uncertainty caused by different GCMs on the potential distribution areas of *A. inebrians*, respectively. We did not directly adopt the ensembles of the GCMs as in previous studies [48] but used the average of the results predicted under four GCMs. The average results not only mitigate the effects of future climate uncertainties by GCMs, but also preserve the impact of the spatial pattern of each GCM on the final results. Furthermore, three RCPs were also used to explore the impact of climate uncertainty caused by different $CO_2$ emissions on the potential distribution areas of *A. inebrians*. It has been reported that the maximum possibility of $CO_2$ emission scenarios in China is RCP4.5 in the future [62]. Our average results under RCP4.5 indicate that the adaptive regions of *A. inebrians* in 2050 are significantly greater than currently observed, mainly distributed in central Inner Mongolia, southern Gansu, Ningxia, eastern Inner Mongolia, Yunnan, most parts of Qinghai, Shaanxi, and Sichuan. However, the changes in the adaptive regions are not significantly different in 2070, with only small plaque growth in Southeast China and sporadic reductions in Shaanxi.

Samples and environmental variables are two important factors in ecological niche modeling, while sample bias and different strategies for selecting environmental variables can also seriously influence the results of ecological niche modeling [63–65]. In our study, the sample bias was reduced by utilizing the nearest neighbor method (i.e., randomly removing one of the two points below the minimum neighbor distance) [66]. At the same time, principal component analysis and correlation analysis were used to select the environmental variables used. Based on the processing of samples and rational selection of environmental variables, the ecological niche model obtained good prediction results, which reinforces the reliability of our results. Moreover, it is important to note that the above estimation of the potential distribution regions of *A. inebrians* was only based on Maxent. However, the ENM alone is not successful at predicting the eventual spread of a species [67], many factors other than climate, such as population processes, biotic interactions, dispersal ability, interactions between demographic, and landscape dynamics, also play an important part in determining species distributions [68,69]. Furthermore, land use patterns may play an important role in predicting the potential distributions [70]. Therefore, a comprehensive model combined with all the above mentioned factors is necessary for the prediction of species-specific responses to climate change and useful agricultural suggestions to the managers and administrators.

**Author Contributions:** Conceptualization, W.-T.W. and L.H.; methodology, L.J.; software, J.-M.J.; formal analysis, J.-M.J. and W.-T.W.; data curation, J.-M.J.; writing—original draft preparation, J.-M.J.; writing—review and editing, L.J., L.H. and W.-T.W.; funding acquisition, L.H. and W.-T.W. All authors have read and agreed to the published version of the manuscript.

**Funding:** This work was supported by the National Natural Science Foundation of China (No. 31560127; 41977420), Natural Science Foundation of Gansu Province (No. 21JR11RA023), Gansu Provincial First-class Discipline Program of Northwest Minzu University (No. 11080305; 41977420; 41671076), the Research Fund for Humanities and social sciences of the Ministry of Education (No. 20XJAZH006), Key Research and Development Program of Ningxia Hui Autonomous Region (No. 2021BEG02009), and the innovation team of intelligent computing and dynamical system analysis and application.

**Institutional Review Board Statement:** Not applicable.

**Informed Consent Statement:** Not applicable.

**Data Availability Statement:** Not applicable.

**Acknowledgments:** The authors are grateful to the anonymous reviewers for their constructive suggestions and comments that have helped to improve the quality of this paper.

**Conflicts of Interest:** The authors declare no conflict of interest.

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
