# Peer review of "The Future Climate under Different CO2 Emission Scenarios Significantly Influences the Potential Distribution of Achnatherum inebrians in China"

_sustainability, doi:10.3390/su14084806_

Round 1

Reviewer 1 Report

Dear authors,

I trully appreciate your research work. 

  1. In "Introduction", you have a PCR instead of RCP. Please check the entire paper for similar issues.
  2. Formulas don't have numbers, and their explanations in the text aren't so clear. Only as a suggestion, you can use: ", where: -; -; -."
  3. (Fig 3,5a) is mentioned several times, but it's not clear what do you mean - Fig. 3 and Fig. 5a?
  4. In the same time, presenting Fig. 3-6 in a row, are making hard to identify the meaning of each graph. Also as a suggestion, each figure could be inserted after the paragraph that mentions it.
  5. Please check "References". Not all the mentioned papers are written the same - see the caps vs. small letters, punctuation marks and so on.

Best regards and good luck in your future activity!

Author Response

1. Comment: In "Introduction", you have a PCR instead of RCP. Please check the entire paper for similar issues.

Response: We have changed “PCR” to “RCP” in the highlights of the fifth paragraph of the Introduction. Thanks.

2. Comment: Formulas don't have numbers, and their explanations in the text aren't so clear. Only as a suggestion, you can use: ", where: -; -; -."

Response: Thank you for your suggestion. We have added the numbers of the formulas, and explained the formulas in detail.

3. Comment: (Fig 3,5a) is mentioned several times, but it's not clear what do you mean - Fig. 3 and Fig. 5a?

Response: Fig 3,5a means Fig. 3 and Fig. 5a. We have revised that.

4. Comment: In the same time, presenting Fig. 3-6 in a row, are making hard to identify the meaning of each graph. Also as a suggestion, each figure could be inserted after the paragraph that mentions it.

Response: Thank you for your suggestion. We retyped the position of Fig. 3-6.

5. Comment: Please check "References". Not all the mentioned papers are written the same - see the caps vs. small letters, punctuation marks and so on.

We have checked all references, and highlight the changes in yellow.

Reviewer 2 Report

This paper deals with the influence of the climate change driven by CO2 emissions on the potential distributions of A. Inebrians, a poisonous weed that threats livestock. The paper used database of the natural distributions of the A. Inebrians to identify suitable conditions for the plant. A suite of ecological (Ecological Niche Model) and climate models were used to predict future spatial distributions of the weed, in 2050 and 2070.

In terms of the influence of the climate, they found that, in 2050, the potential distribution is higher with a lower volume of precipitation during the coldest quarter. In terms of the influence of CO2 increase, three from the four models used predict increase of the distribution of A Inebrians with the increase of CO2. One model predict the contrary, i.e. low CO2 emissions scenarios are more suitable for the expansion of A Inebrians. However, in general, the results reveal that the increase of CO2 emissions tend to increase the A Inebrians distributions, at least in 2050

At this point, is not clear for me if the influence of the CO2 in the weed distribution occurs directly - by an increase in the photosynthesis rate (e.g., CO2 fertilization), or indirect, by changing the climate conditions that, in turn, influence the plant distribution. I think is important to clarify.

Author Response

Comment: At this point, is not clear for me if the influence of the CO2 in the weed distribution occurs directly - by an increase in the photosynthesis rate (e.g., CO2 fertilization), or indirect, by changing the climate conditions that, in turn, influence the plant distribution. I think is important to clarify.

Response: Special thanks to you for your good comments. We have reorganized the writing of our manuscript to indicate that CO2 indirectly affects the potential future distribution of A. inebrians through changing the climate conditions. The relevant revisions are detailed in the highlights of the Abstract and the first and second paragraphs of the Introduction. In addition, we have described the figures in the results in detail.

Reviewer 3 Report

General comment: The current study entitled CO2 emissions scenarios significantly influence the potential distribution of Achnatherum inebrians in China has good professional quality. This study has modelled the effect of three different CO2 emission scenarios (RCP 2.6, 4.5 and 8.5) on the potential future distributions of Achnatherum inebrians. The title is not expressive, it gives a result, which is not usual in a scientific article. I suggest that the title should rather indicate the subject of the study. The scenario itself does not affect the spread of weeds, only the associated change in CO2 concentration. The wording of the title is incorrect. It is recommended to be revised.

Abstract: The structure of this part of the manuscript is proper and reflects well the relevance and the added value of the research project.

Introduction: Well-structured, with good professional quality. The objectives are correctly formulated, expressing the research much better than the title. Links do not need to be superscripted.

Material and Methods: Well-detailed, but there was a mistake in the placement of the tables and formulas.

Results: This part is also good, although the placement of the figures is not good, but I guess it will be reedited. It is already sufficiently emphasised here that they are talking about POTENTIAL spread. They rightly mention that changes in temperature and rainfall are also strongly influencing factors. Their relative contribution should be compared to that of CO2 concentration. Figures 3 and 4 need to be reconsidered. They are not interpretable in their present form.

Discussion: Well-detailed. Here, too, they emphasise the complex effect of precipitation and temperature in addition to CO2. Perhaps this complexity should also be referred to in the title. However, I miss the practical approach that if the increase in CO2 concentrations is human induced, how can weed spreading be affected by agricultural activity? Alternatively, do other plant species come into play as competing species that could affect the spread? So, the complexity should be assessed and evaluated in higher depth, and not simplified as the title suggests.

Author Response

1. Comment: The title is not expressive, it gives a result, which is not usual in a scientific article. I suggest that the title should rather indicate the subject of the study. The scenario itself does not affect the spread of weeds, only the associated change in CO2concentration. The wording of the title is incorrect. It is recommended to be revised.

Response: Thank you for your suggestion. We have changed the title to “Future climate under different CO2 emissions scenarios significantly influence the potential distribution of Achnatherum inebrians in China”.

2. Comment: Introduction: Links do not need to be superscripted.

Response: We have changed all links. Thank you.

3. Comment: Material and Methods: Well-detailed, but there was a mistake in the placement of the tables and formulas.

Response: We have reedited the tables and formulas.

4. Comment: Results: This part is also good, although the placement of the figures is not good, but I guess it will be reedited.

Response: We have reedited the placement of the figures.

5. Comment: It is already sufficiently emphasised here that they are talking about POTENTIAL spread. They rightly mention that changes in temperature and rainfall are also strongly influencing factors. Their relative contribution should be compared to that of COconcentration.

Response: We have reorganized the writing of our paper to disambiguate expressions as much as possible. Our research focuses on the impact of future climate change on the potential distribution of A. inebrians. And future climate mainly determined by COemission concentrations. That is, COemission concentrations indirectly affect the potential future distribution of A. inebrians through changing the climate conditions. Therefore, we didn’t compare relative contribution between COconcentration and climate (i.e. temperature and rainfall).

6. Comment: Figures 3 and 4 need to be reconsidered. They are not interpretable in their present form.

Response: Thank you for your comments. We have illustrated Figures 3 and 4 in detail.

7. Comment: Discussion: Well-detailed. Here, too, they emphasise the complex effect of precipitation and temperature in addition to CO2. Perhaps this complexity should also be referred to in the title. However, I miss the practical approach that if the increase in CO2concentrations is human induced, how can weed spreading be affected by agricultural activity? Alternatively, do other plant species come into play as competing species that could affect the spread? So, the complexity should be assessed and evaluated in higher depth, and not simplified as the title suggests.

Response: We have reworked the logic of the Discussion section. We focus on the differences in the potential distribution of A. inebrians under future climate of different CO2 emission scenarios. Different CO2 emissions will determine different future climates. And the direct factor affecting species distribution is climate, the indirect factor is CO2 emission scenarios. In addition, we did not consider dispersal and inter-species interaction in the potential distribution of A. inebrians simulated by the ecological niche model. This is a shortcoming of our study, which we also stated in the last paragraph of the Discussion.
